**METHOD**

# RobusTAD: reference panel based annotation of nested topologically associating domains

Yanlin Zhang[1], Rola Dali[1] and Mathieu Blanchette[1]*

*Correspondence:
blanchem@cs.mcgill.ca

[1] School of Computer Science, Mcgill University, Montréal, Canada

**Abstract**

Topologically associating domains (TADs) are fundamental units of 3D genomes and play essential roles in gene regulation. Hi-C data suggests a hierarchical organization of TADs. Accurately annotating nested TADs from Hi-C data remains challenging, both in terms of the precise identification of boundaries and the correct inference of hierarchies. While domain boundary is relatively well conserved across cells, few approaches have taken advantage of this fact. Here, we present RobusTAD to annotate TAD hierarchies. It incorporates additional Hi-C data to refine boundaries annotated from the study sample. RobusTAD outperforms existing tools at boundary and domain annotation across several benchmarking tasks.

**Keywords:** Hi-C, TAD, Nonparametric test, Dynamic programming

## Background

The hierarchical organization of mammalian chromosomes within the nucleus has been increasingly identified as an essential factor for cellular functions such as gene regulation, cell fate determination, and evolution [1]. Though the multi-scale chromosomal folding revealed by chromosome conformation capture techniques—such as Hi-C—are frequently studied [2–5], understanding its structural and functional roles is still in its infancy. Moreover, identifying spatial elements such as TADs and loops from Hi-C data is challenging, particularly due to the relatively low resolution permitted by Hi-C datasets of typical sequencing depth (200–500M valid read pairs) [6, 7].

TADs are self-interacting regions along the chromosome, manifesting as squares along the diagonal of Hi-C contact maps [8]. Researchers used to perform TAD annotations from Hi-C datasets at low resolution (i.e., 40 kb), and define TADs as megabase-scale structural elements [5]. Later, researchers observed that TADs can be much smaller (i.e., tens to hundreds kilobases) in mammalian chromosomes by investigating high-coverage and high-resolution Hi-C contact maps [3]. This observation led researchers to detect

TADs at high resolutions [1, 8, 9]. In addition, the study of sequence-encoded factors, such as CTCF (CCCTC-binding factor), that influence TAD formation also requires researchers to annotate TADs at much higher resolutions. Meanwhile, TAD-within-TAD (or subTAD) organization also gathered significant attention in recent years [8].

Many computational tools for TAD annotation have been proposed [3, 5, 10–23]. These approaches can be classified as either one-dimensional (1D) score-based and or matrix-based approaches. Score-based approaches, such as TopDom [10], Insulation Score (IS) [22], and OnTAD [15], assign each locus a score representing the strength of a potential TAD boundary and subsequently detect TAD boundaries by identifying local optima among the list of scores. Matrix-bases approaches directly utilize two-dimensional (2D) data instead of transforming it into a 1D statistic. For example, Arrowhead [3] transforms the Hi-C contact map into an arrowhead-shaped feature map and subsequently identifies TADs by searching for corners in the transformed matrix. Chen and colleagues formulated TAD annotation as graph segmentation [23], viewing a Hi-C contact map as an adjacency matrix to model a chromosome as a graph and identified TADs through graph Laplacians.

Despite the numerous TAD annotation tools available, the identification of TAD hierarchy and the precise location of TAD boundaries at high resolution remains challenging. Most TAD annotation tools are designed for high-coverage contact maps, or operate at low-resolution. As reviewed in previous studies [7, 24–26], these tools are not robust to resolutions and sequencing coverages. Additionally, the predictions of TADs and domain boundaries show limited agreement across tools. Finally, most existing algorithms only use the contact map of the sample of interest to annotate TADs. Their performance is thus limited by the sequencing depth of the Hi-C data from that study. Most Hi-C data sets produced to date are in the range of 200M to 500M valid read pairs, with only a slow increase over time. We anticipate that this situation remains until sequencing costs drop dramatically.

The issue of insufficient data coverage also exists in many other biological data analysis tasks. Researchers often address this issue through introducing data from samples other than the sample under study [27, 28]. For example, though a SNP array typically covers only a few hundreds of thousands of loci, it is routine to infer unobserved genotypes through imputation with a reference panel containing a larger spectrum of genotyped variants [29, 30]. Similarly, homology modeling for protein structure prediction exploits a database of known structures [27]. Conversely, the vast amount of published Hi-C datasets are seldom employed to annotate TAD. While some research suggests that certain TADs are cell type or even replicate-specific[31], there are also observations demonstrating that many domain boundaries, or even entire TADs, are conserved across different cell types [32]. We hypothesize that since TADs are determined by genome sequence, epigenetics, and cellular dynamics, the vast majority of TADs present in a given cell-type/condition are also present in multiple other samples. Given the large number of Hi-C datasets in public depositories (e.g., [33]), the stage is set to develop TAD annotation approaches that better utilize existing Hi-C data sets. Recently, we introduced RefHiC [34], a reference panel enabled approach for TADs and chromatin loops annotation. Although we demonstrated that the introduction of a Hi-C reference panel enables RefHiC to significantly outperform alternatives in TAD annotation,

RefHiC suffers from several limitations: (i) RefHiC does not predict TAD hierarchies; (ii) the computationally intensive process of projecting Hi-C samples onto the latent space impedes including more samples into the reference panel.

Here, we introduce RobusTAD. RobusTAD is a TAD annotation algorithm that provides accurate and robust TAD annotation at high resolution. It improves TAD boundary annotation by leveraging publicly available Hi-C data and achieves superior performance by exploiting locally matched chromosome conformations (LMCC). Following TAD boundary annotation, it uses non-parametric tests and a dynamic programming algorithm to obtain the optimal nested TAD structure. RobusTAD outperforms existing TAD callers in a variety of contexts. We further demonstrated that RobusTAD is robust to low sequencing coverage and can produce high-resolution TAD annotations from Hi-C data of typical sequencing depth (250–300M reads). Finally, we show that applying RobusTAD to predict TAD at high resolution facilitates dissecting TADs according to transcription factor binding site profiles around TAD boundaries and consequently probe TAD formation.

## Results

### Overview of RobusTAD

RobusTAD takes a normalized Hi-C contact matrix as input and calls TADs in three steps (Fig. 1): (i) Low-accuracy TAD boundary identification based on the study sample; (ii) Refinement of TAD boundary locations based on locally-matched chromosome conformations from a reference panel of Hi-C data sets; (iii) Pairing of left and right boundaries into an optimal nested domain hierarchy.

Study sample based boundary identification is based on seeking local maxima in a vector of 1D nonparametric TAD boundary scores. RobusTAD assigns separate left and right TAD boundary scores to each locus by performing genomic distance stratified rank-sum test between upstream/downstream inter- and intra-domain interactions.

RobusTAD refines boundary calls made on the study sample by utilizing selected Hi-C samples from a reference panel. For a given candidate TAD boundary at position $p$, we define locally matched chromosome conformations $LMCC(p)$ as a collection of Hi-C samples in which a predicted TAD boundary occurs within 25 kb (i.e., five 5-kb bins) of $p$. It then computes refined boundary scores for the 50 kb region by combining the boundary scores from $LMCC(p)$ and the study sample itself; the position that reaches the maximum score is the final high-resolution boundary prediction.

In the third step, RobusTAD assembles a hierarchy of nested domains by pairing left and right boundary candidates using a dynamic programming algorithm, inspired by the Nussinov algorithm for RNA secondary structure prediction [35], that maximizes the full chromosomal TAD score (i.e., the sum of TADs scores). RobusTAD computes the TAD score with the distance-stratified rank-sum statistic of interactions between intra- and inter-domain in both upstream and downstream. To avoid the TAD score inflation caused by the presence of sub-TADs within a given region, lower-level domains previously identified during the execution of the algorithm are excluded from a region's TAD score calculation. The algorithm is guaranteed to produce the globally optimal nested TAD hierarchy.

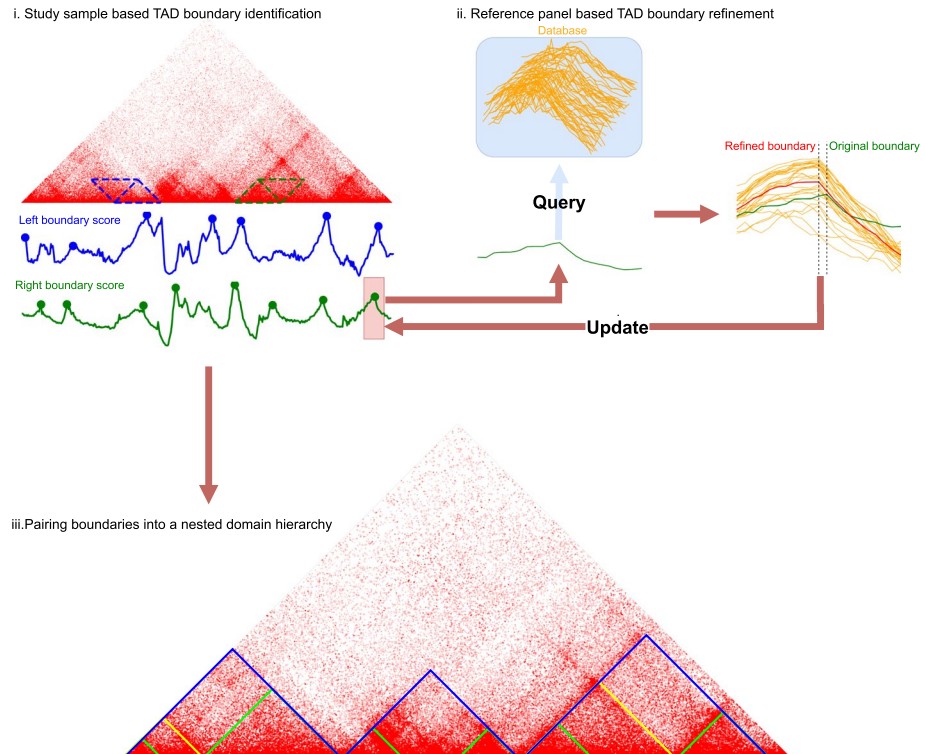

**Fig. 1** Overview of the RobusTAD algorithm. RobusTAD detects TAD boundaries in three steps (i, ii, and iii). First, approximate left and right TAD boundaries are identified based on the study sample. Second, RobusTAD identifies locally matched chromosomal conformations (LMCCs) from a panel of reference data sets, and uses those LMCCs to refine the position of each TAD boundary. Finally (step iii), refined left and right boundaries are paired to form an optimal nested TAD hierarchy

## Comparison with existing TAD callers

We compared the performance of RobusTAD to 14 other TAD callers: TopDom [10], Armatus [11], deDoc [12], Arrowhead [3], HiTAD [13], EAST [14], OnTAD [15], CaTCH [16], Grinch [17], Domaincall [5], GMAP [18], HiCSeg [19], RefHiC [34], and IC-Finder [20]. We tried to include hierarchical TAD callers TADtree [36], TADpole [25], and SuperTAD [37] in our benchmarking, but they could not complete within a week of running time and hence we had to exclude them. Since RobusTAD and some other TAD callers detect nested TADs, we define TADs that do not contain any smaller TADs as level 0 TADs, and TADs that contain one or multiple smaller TADs as level 1+ TADs. We performed the benchmark evaluation experiments proposed by Zufferey et al. [24] on chromosomes 15–17 of Hi-C data for human GM12878 cells [3], down-sampled to 250 Million valid read pairs. We conducted all studies at 5 kb resolution and employed iterative normalized [38, 39] Hi-C contact maps as input. The running time varies significantly among tools in annotating TADs from all three chromosomes. OnTAD and callers that do not produce nested TADs are able to annotate the three chromosomes within 20 min. RobusTAD took a total of 1.5 hours to annotate the three chromosomes. Most of other nested TAD callers also require a similar amount of time.

We first compared the number and size of TADs identified by each tool. Interestingly, the number of TADs varies greatly, with Arrowhead identifying less than 500

TADs and OnTAD identifying around 3800. Most of the tools, including RobusTAD identified $1000 - 2000$ TADs (Fig. 2a). Tools that identify more TADs naturally produce smaller TADs (Fig. 2b). RobusTAD identified TADs of a wide range of sizes, with a median size of 170 kb. UMAP embedding [40] on Measure of Concordance (MoC) [24] values among pairwise callers identifies three major caller groups. Among all callers, RobusTAD, RefHiC, Domaincall, GMAP, Armatus and HiTAD form a cluster with an average within-cluster MoC of 0.47 (Fig. 2c).

We then examined the quality of the TAD annotations produced by each tool. Additional file 1: Fig. S1 shows an example genomic region (chr16:10.2–12 Mb), with TADs annotated by RobusTAD and other TAD callers. TADs lack ground-truth annotation, so it is impossible to calculate the accuracy of TAD predictions. Thus, we used three metrics to evaluate each predicted TAD's quality. (i) RobusTAD's TAD

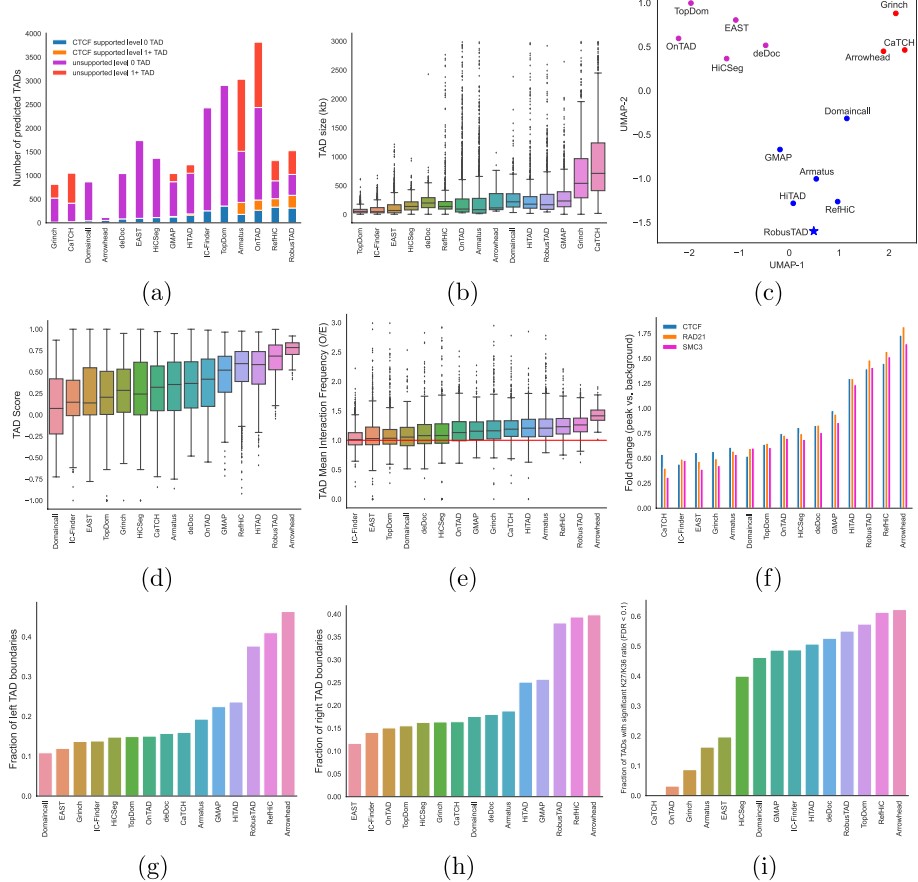

**Fig. 2** Comparison of RobusTAD, and 14 other TAD callers on a GM12878 Hi-C data set of 250M valid read pairs. **a** Number of TADs predicted by different tools, and proportion of predicted TAD boundary pairs that are supported by CTCF ChIA-PET data. **b** Size distribution of predicted TADs. **c** U-MAP analysis performed on the Pearson's correlation matrix of the matrix of pairwise MoC between TADs identified by all callers. Comparison of the quality of TADs predicted by different tools using RobusTAD's TAD score (**d**), and TAD mean interaction frequency (observed/expected) (**e**). **f** Fold change of structural protein peak signals at TAD boundaries for CTCF, RAD21, and SMC3. Number of left (**g**) and right (**h**) boundaries that contain at least one CTCF ChIP-seq peak. **i** Fraction of TADs with significant log10 ratio between H3 K27 me3 and H3 K36 me3. Note: Panels **c**, **f–i** are generated with benchmarking code created by Zufferey et al. [24]

score (see the "Methods" section, on the full coverage data), (ii) mean interaction frequency (Observed over Expected, on the full coverage data) inside a TAD, and (iii) agreement with CTCF ChIA-PET data. The TAD score measures the enrichment of interaction frequencies inside a TAD by using its neighboring regions as the background. It ranges from $-1$ to 1; positive values indicate higher interactions within the TAD than across its boundaries. RobusTAD ranks second based on mean TAD score (Fig. 2d); only Arrowhead, a tool that predicts approximately 5 times fewer TADs, reaches a higher mean TAD score. Similar results are obtained when assessing predicted TADs based on average observed/expected ratios (Fig. 2e).

Loop TADs in mammalian genomes are TADs that exhibit a strong contact between their boundary loci [8]. We assessed TAD annotations by comparing predicted TAD boundary pairs with CTCF ChIA-PET data (allowing up to one 5-kb bin mismatch). Figure 2a shows that 582 (38%) TAD predictions made by RobusTAD match ChIA-PET data. This is both the largest number and the largest proportion of supported TAD predictions across all tools.

We also studied the performance of TAD annotations at the level of individual TAD boundaries. We first calculated the enrichment for ChIP-Seq signals of structural proteins (CTCF, RAD21, and SMC3) associated with predicted TAD boundaries (Fig. 2f and Additional file 1: Fig. S2). TAD boundaries predicted by most tools are enriched for these architectural proteins. RobusTAD ranked $3^{rd}$ for the mean fold-change enrichment of the three structural proteins. Figure 2g, h compare left and right boundaries to CTCF ChIP-seq data (allowing 1-bin mismatch) separately; RobusTAD ranked $3^{rd}$ for both left and right boundary predictions, slightly outperformed by Arrowhead, and the other reference panel enabled tool, RefHiC. Histone marks usually correlate with regulatory activity, and TADs are typically consistently enriched for either activating (H3 K36 me3) or repressive (H3 K27 me3) marks. We calculated the ratio between H3 K27 me3 and H3 K36 me3 within each TAD prediction and counted the fraction of TAD predictions where this ratio was particularly large or small (see the "Methods" section). Figure 2i shows RobusTAD ranked $4^{th}$, slightly outperformed by TopDom, RefHiC, and ArrowHead.

We then conducted a visual examination of TAD predictions made by all tools. We performed this analysis using rescaled pileup plots generated by Coolpup.py [41], with the "--local" and "--rescale" options. Figure 3 and S3 show most tools, including RobusTAD, identified TADs as square regions with increased interaction frequency, dot-corners, and well-defined domain borders. In contrast, tools such as Domaincall, deDoc, and Armatus yielded TAD predictions with less distinct domain borders. TAD predicted by Grinch and HiCSeg are less enriched for Hi-C contacts. In addition, we observed clear vertical and horizontal stripes with increased interaction frequencies at the boundaries of TADs predicted by RobusTAD, HiCTAD, RefHiC, TopDom, and GMAP. The two stripes indicate these TAD callers can identify both TADs and sub-TADs.

Taken collectively, these results suggest that RobusTAD is the most accurate TAD caller, as it is the only tool ranking among the top four TAD callers in all accuracy metrics.

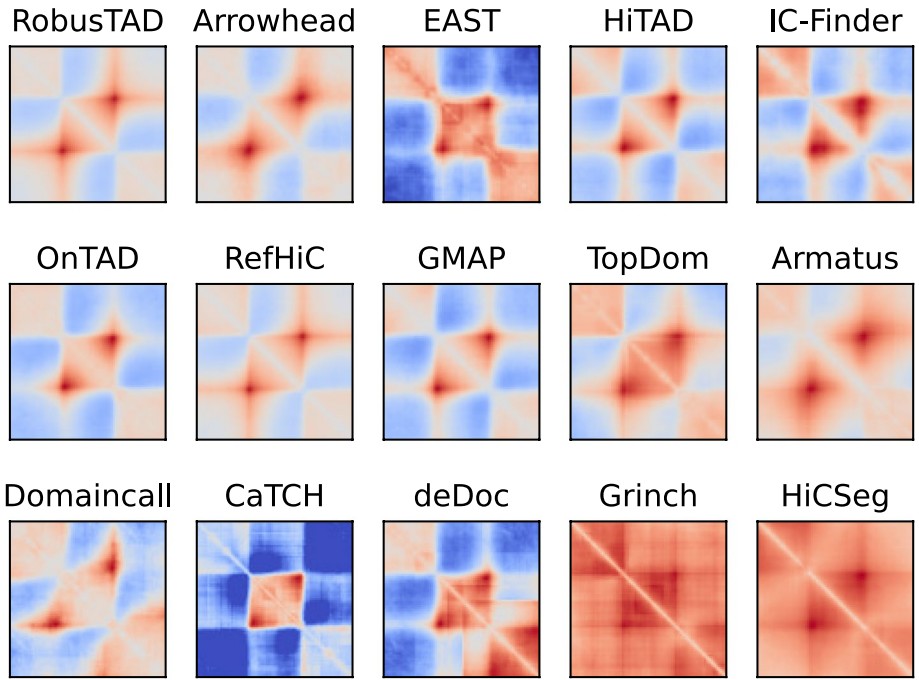

**Fig. 3** Visual comparison of TADs predicted by RobusTAD and 14 other tools from GM12878 Hi-C data. The rescaled pileup plots aggregate areas around TAD predictions in the full-coverage Hi-C contact map. TAD predictions were annotated against a downsampled Hi-C contact map containing 250M valid read pairs

### RobusTAD is robust to low sequencing coverage Hi-C data

TAD annotation is typically sensitive to sequencing depth. Many TAD callers do not perform well when the sequencing depth is low, and boundaries detected from contact maps of differing sequencing depths have been reported to lack reproducibility [6, 7]. We evaluated how RobusTAD and other tools performed on Hi-C contact maps of varying sequencing depths (generated from the combined Hi-C data set for GM12878 [3]), from data set of 4B valid read pairs down to downsampled versions with as few as 62.5M valid read pairs. As illustrated in Fig. 4a, different tools react differently to reduced coverage: some (including RobusTAD) conservatively reduce their predictions, while others are unaffected or even increase their number of predictions. These results suggest that tools like RobusTAD can mitigate false-positive identifications effectively.

We then assessed the tool's robustness by measuring the similarity between the predictions made on the highest coverage data set (i.e., 4 billion valid read pairs) to those made on downsampled data, both at the levels of TAD boundaries (Fig. 4b, using the Jaccard index) and TADs (Fig. 4c, using the Measure of Concordance). RobusTAD and RefHiC, the two reference panel based approaches, exhibit the highest levels of consistency at TAD boundaries level. Following Zufferey et al., we used the Measure of Concordance (MoC) [24] to compare two sets of TAD predictions. MoC does not handle overlapping TADs, thus we only included TADs that do not include any smaller TADs in this analysis. Figure 4c shows RefHiC, HiCSeg, and RobusTAD outperformed other tools at most levels of coverage. Last, we evaluated TAD and domain boundary prediction accuracy by comparing predictions to CTCF ChIA-PET (Fig. 4d) and CTCF ChIP-Seq (Additional file 1: Fig. S4, S5) data. For contact maps containing more than 250M valid read pairs,

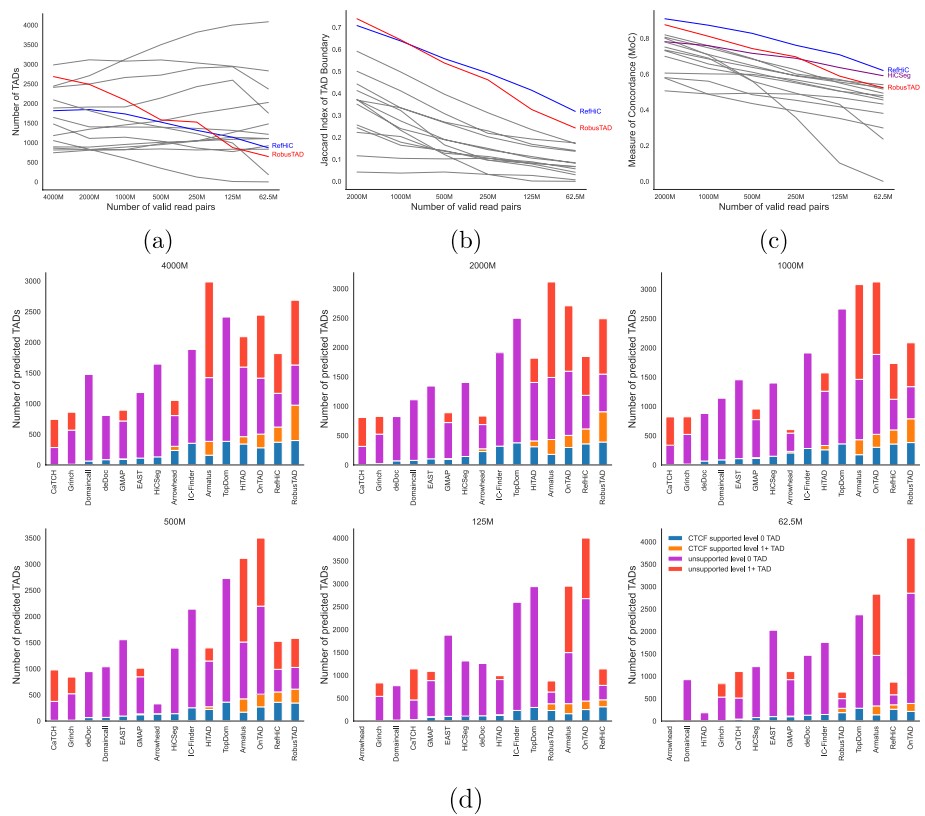

**Fig. 4** Comparison of RobusTAD and other 14 TAD callers on downsampled GM12878 Hi-C data. **a** Number of TAD predicted from downsampled Hi-C data. Jaccard index of predicted TAD boundaries (**b**) and Concordance between TADs (**c**) predicted on full data (4B valid read pairs) compared to those predicted on the downsampled Hi-C data. **d** Number of TADs predicted from downsampled data, and proportion of predicted TAD boundary pairs that are supported by CTCF ChIA-PET data

RobusTAD performed the best. On data containing fewer valid contact pairs, RobusTAD is only slightly outmatched by RefHiC. Although OnTAD, Armatus, and TopDom also identified more CTCF supported TADs from data containing few valid read pairs, they were less accurate than our tools, as they identified many more TADs that were not supported by CTCF data.

## RobusTAD performs well across cell types

Here, we demonstrate that RobusTAD performs well across cell types. We applied RobusTAD and five other TAD callers (GMAP, HiTAD, Arrowhead, OnTAD, and Ref-HiC), to annotate TADs from Hi-C contact maps derived from IMR-90 and K562 cell lines [3]. The rescaled pileup plots show that all tools successfully identified TADs as squares with increased interaction frequencies and dot-corners (Fig. 5a). In addition, TAD predictions made by all tools contain vertical and horizontal stripes with increased interaction frequency at its boundary locations. These stripes indicate these tools accurately detect TADs and subTADs from Hi-C contact maps. The number of TAD predicted by different tools from the two contact maps ranges from 500 to 3500, with RobusTAD detecting 2630 and 1710 TADs from the two contact maps (Fig. 5b). The mean expectation normalized interaction frequency (O/E) within a TAD further

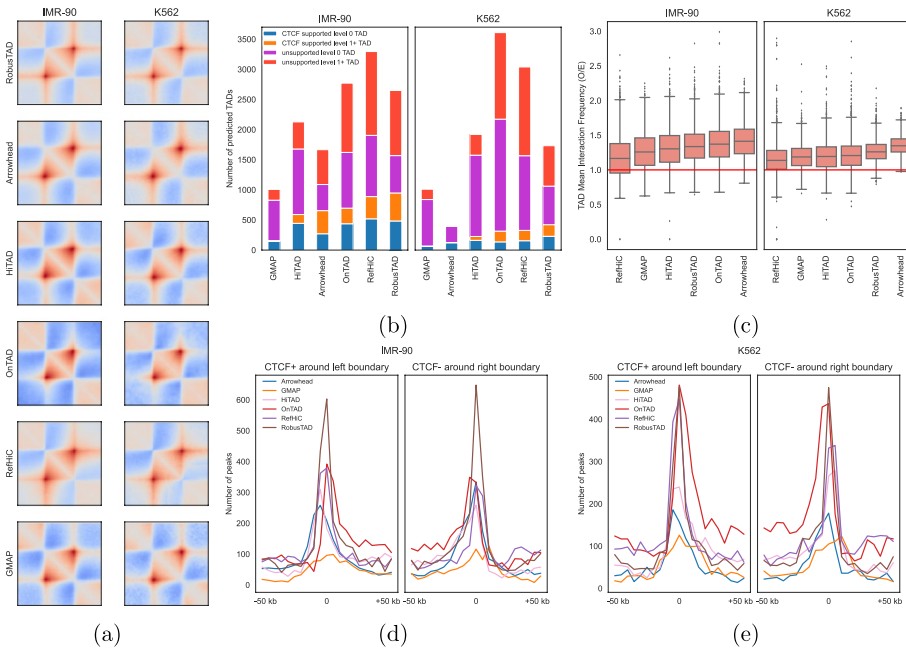

**Fig. 5** Comparison of RobusTAD and other five TAD callers on Hi-C data derived from IMR- 90 and K562 cell lines. **a** Rescaled pileup plots over predicted domains. **b** Number of TADs predicted by different tools, and proportion of predicted TAD boundary pairs that are supported by CTCF ChIA-PET data. **c** TAD mean interaction frequency (observed/expected). Occupancy of ChIP-seq identified forward and reverse CTCF binding sites as a function of distance to left (**d**) and right (**e**) boundary annotations

confirms that all tools, including RobusTAD, successfully identified TADs as a region with increased *cis*-contact pairs (Fig. 5c). Next, we compared boundary pairs to CTCF ChIA-PET data (Fig. 5b). The ChIA-PET data for IMR- 90 contains 4957 contact pairs and the ChIA-PET data for K562 contains 2168 contact pairs. RobusTAD identified the most ChIA-PET supported TADs from both contact maps. Last, we evaluated boundary prediction accuracy by comparing predicted boundary to forward and reverse CTCF binding sites identified by ChIP-Seq experiment. Left and right boundaries predicted by RobusTAD are more enriched by CTCF binding sites than boundaries identified by alternative tools (Fig. 5d, e).

### RobusTAD reveals multiple types of TADs

Building upon the high accuracy and resolution of RobusTAD, we used it to perform a study of TADs functionalities. We focused on TAD predictions made on the full set of autosomes for a combined Hi-C data set obtained from the GM12878 cell line [3]. We characterized a TAD as a binary vector of dimension $2 \times 116 = 232$, representing the ChIP-seq derived occupancy of 116 transcription factors [42] at its left and right domain boundaries. Unlike previous work that analyzed regulatory elements based on transcription factor binding, such as chromatin state annotations, this paired analysis of boundary regions offers a unique perspective often missing in earlier studies [43]. We identified six TAD groups by applying the UMAP algorithm [40] to project TADs onto a 2D space, followed by *K*-means clustering [44] (Fig. 6a). The symmetric pattern observed in the UMAP projection (Fig. 6a) and the group-averaged occupancy vectors (Fig. 6b)

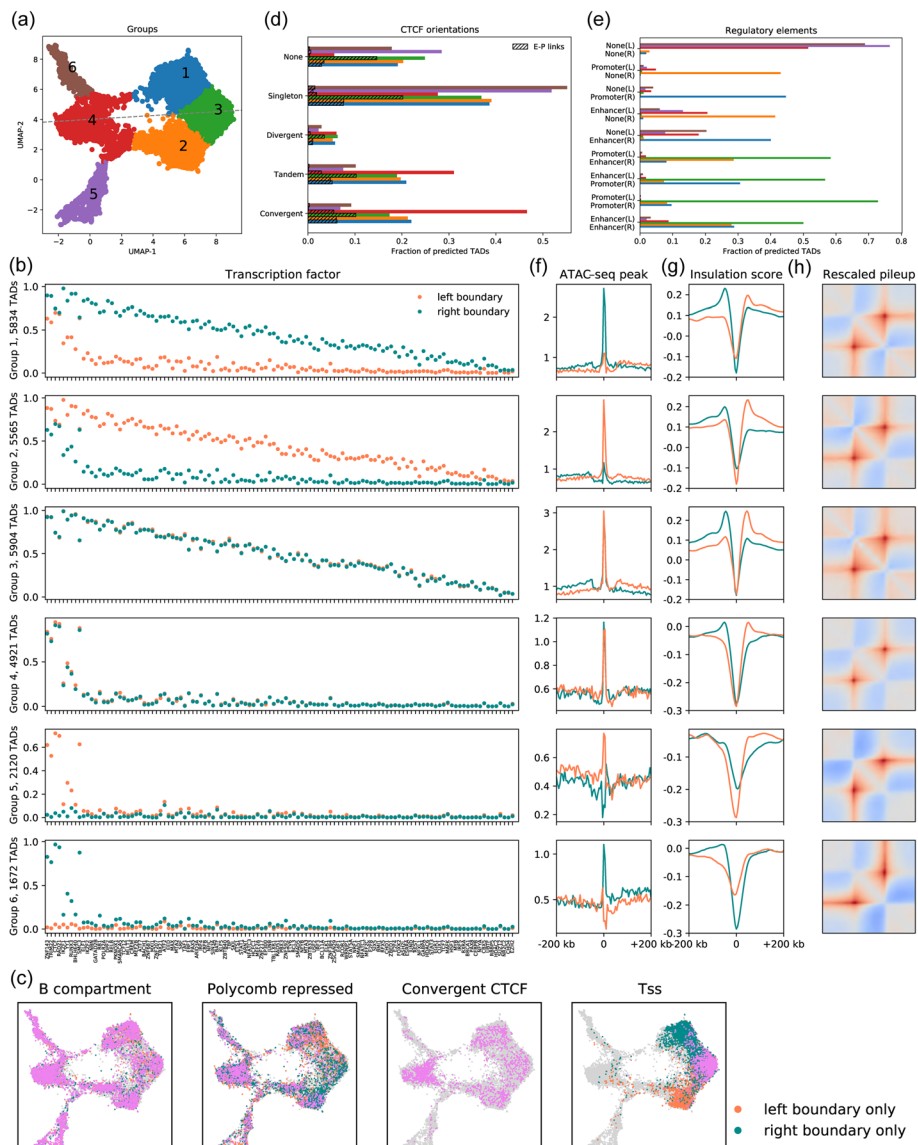

**Fig. 6** Applying RobusTAD to Hi-C data for GM12878 cells reveals TAD groups. **a** A two-dimensional UMAP projection of TADs based on the occupancy of transcription factors at domain boundaries. **b** Occupancy of transcription factors in each group of TADs. **c** two-dimensional distributions in the UMAP projected space of TADs associated with different features. **d** Occupancy of different pairs of directional CTCF binding sites at domain boundaries. E-P links are domains supported by POLR2 A ChiA-PET data. **e** Proportion of annotated TADs with different regulatory element combinations at domain boundaries. Enrichment of ATAC-seq peaks (**f**) and insulation scores (**g**) at domain boundaries for each TAD group. **h** Rescaled pileup plots over TAD predictions for each TAD group

indicate that left and right domain boundaries play similar functional roles. Chromatin structural proteins such as ZNF143, CTCF, YY1, and subunits of cohesin complex (SMC3, and RAD21) are the most enriched proteins in all six groups. We also observed TRIM22 highly enriched at domain boundaries in all groups (Fig. 6b).

The distribution of transcription factors at TAD boundaries (Fig. 6b) and the chromatin accessibility quantified as the average count of ATAC-seq peaks at domain

boundaries (Fig. 6f) motivate us to interpret group assignments by investigating transcriptional activity and chromatin accessibility.

Groups 3 is characterized by having both boundaries exhibiting evidence of transcriptional activity, with high TF occupancy (Fig. 6b) and chromatin accessibility (Fig. 6f)), often involving pairs of regions annotated as active enhancers or promoters (as annotated by ENCODE's combined Segway ChromHMM segmentations [42]) (Fig. 6c, e). This is confirmed this by comparing boundary pairs to Enhancer-Promoter links identified by a POLR2 A ChiA-PET experiment [42] (Fig. 6d). Additionally, we observe that Enhancer-Promoter pairs can mediate TAD formation even in the absence of CTCF binding sites, with 14.5% of Group- 3 TADs corresponding to such links but lacking CTCF occupancy.

Groups 1 and 2 only display evidence of transcriptional activity at one of their two boundaries, with the inactive boundary showing reduced TF occupancy. Although those three groups are quite different in terms of their activity profiles, their insulation profiles (Fig. 6g) and pile-up plots (Fig. 6h) are nearly identical.

Groups 4, 5, and 6 are characterized by TADs whose both boundaries are located in repressive chromatin (Fig. 6c) with low TF occupancy (Fig. 6b) and chromatin accessibility (Fig. 6f), and little overlap with active enhancers/promoters (Fig. 6e). Group- 4 TADs have both boundaries occupied by CTCF and associated structural proteins; these TADs' boundaries also display the highest level of convergent CTCF binding sites (Fig. 6d) and have sharper corner dots than domains in other groups (Additional file 1: Fig. S6), probably because of the reduced level of interactions in surrounding regions. On the contrary, Group 5 and 6 TADs lack CTCF at one or the other of their boundaries. They also exhibit weak insulation scores at the CTCF-free boundary (Fig. 6g, Additional file 1: Fig. S7, S8a), and weak dot corners (Additional file 1: Fig. S6).

Among all groups, domain boundaries in active regions are sharper than in repressed regions (Fig. 6g, Additional file 1: Fig. S7 and S8a). Domains in active regions are more enriched by Hi-C contacts than domains in repressive regions (Additional file 1: Fig. S8b).

We observe that domain boundaries shared by multiple TADs are more enriched for CTCF bindings sites and activated promoters (Additional file 1: Fig. S9). We further studied the hierarchy structure of these TADs by classifying them into singleton TADs (isolated TADs that do not overlap with others), TADs (non-singleton TADs that do not reside within other TADs), and sub-TADs (non-singleton TADs found within larger TADs) (Additional file 1: Note S1 and Fig. S10). We found TADs frequently associated with boundaries marked by convergent CTCF motifs, and subTADs playing an important role in gene regulation, with a substantial portion being E-P links.

## Discussion

Hi-C experiments and their derivatives have become routine in studying 3D genome organization at the genome-wide scale. Many Hi-C studies have been carried out in the past decade, and hundreds of Hi-C datasets have been published. Though this type of data has enabled the discovery of several key levels of 3D genome organizations (e.g., loops, TADs, and compartments), accurately identifying TAD boundaries and the ways in which they assemble to form a TAD hierarchy remain challenging with existing tools,

especially for Hi-C data with typical sequencing coverage. To deepen our understanding of 3D genome organization, high-resolution annotations of TAD hierarchy are required.

RobusTAD annotates high-resolution TAD boundaries and TAD hierarchy from Hi-C contact maps, taking advantage of a reference panel of high-quality Hi-C data sets. Including more reference samples improves annotation accuracy (Additional file 1: Fig. S11 and S12), so one can expect that RobusTAD will continue to get better as its reference panel grows. RobusTAD is based on a novel nonparametric statistic to score both domain boundaries and TADs. It is a distribution-free test to evaluate TAD and TAD boundaries. Thus, it is robust to changes in observation, such as those due to noise and sparsity. In addition, RobusTAD's separate scoring of left and right boundaries eases the dissection of domain boundaries. In contrast, most existing TAD callers do not quantify domain boundaries or amalgamate left and right boundaries as a single insulation locus.

RobusTAD overcomes the statistical challenges caused by high sparsity and signal-to-noise ratio limitation in a Hi-C contact map of typical coverage by identifying and combining many locally matched Hi-C data. At moderate sequencing depths, existing tools often fail at maintaining a low level of false identifications and identify many inaccurate TADs. Although a user can adjust parameters to limit the number of identified TADs in most tools, adjusting parameters is challenging and not necessarily effective at reducing false identification. Within RobusTAD, we use a simple and statistically sound target-decoy search strategy to select TAD boundaries from a list of candidate boundaries. A user only needs to specify the desired false discovery rate (FDR) threshold ($\alpha$) to ensure that the final predictions to contain at most $\alpha$ expected false-positive boundaries.

Our tools outperformed many TAD callers in accuracy and reproducibility in identifying high-resolution TAD from multiple Hi-C contact maps. For instance, both CTCF ChIA-PET and CTCF occupancy data highlight the superiority of RobusTAD at both boundary detection and TAD assembly. The benefits of RobusTAD were shown to be particularly significant in typical moderate-to-low coverage Hi-C data. As demonstrated in Fig. 4a, d−i, RobusTAD can reduce false-positive identifications by identifying slightly fewer TAD boundaries from low-coverage Hi-C contact maps. In contrast, false-positive boundaries increased dramatically in predictions made by most other TAD callers when applied to low-coverage Hi-C contact maps. While RobusTAD was bested by RefHiC in terms of the accuracy of boundary annotation from very low coverage Hi-C data, it outmatched even that tool in terms of predicting CTCF ChIA-PET supported TADs. This advantage of RobusTAD over RefHiC is attributed to the newly developed dynamic programming algorithm for pairing TAD boundaries in RobusTAD. In addition, RobusTAD guarantees full interpretability without a loss of accuracy. When a boundary only appears in the study sample, RobusTAD will usually annotate it without using additional data from its reference panel. LMCC based boundary refinement makes mistake only if the boundary in the study sample is very close (i.e., within 50 kb) to another boundary in the reference panel. Given the superior performance achieved by RobusTAD, we believe this case rarely occurs.

Despite its advantages, RobusTAD has several limitations. First, given RefHiC slightly outperformed RobusTAD in some aspects of TAD boundary annotation, we believe RobusTAD does not fully capitalize on the advantages offered by the reference panel. It might be due to some weak boundaries that cannot be identified in the first step of

RobusTAD, and the heuristic approach for selecting reference samples is efficient but might be not as effective as RefHiC's data-driven approach. We can overcome this by replacing the first step with the deep learning model in RefHiC, at the cost of losing efficiency and interpretability. Second, the dynamic programming is time consuming, in part due to we need to compare two large sets of interaction frequencies to evaluate the score of a large candidate domain. We are planning to improve the running time by sampling a fraction of elements from the two sets to estimate the score of large candidate TADs.

## Conclusions

RobusTAD allows for precise, high-resolution TAD annotation from Hi-C data of a wide range of sequencing depths, all the way down to only 62.5 million contact pairs. RobusTAD improves the performance of TAD boundary annotation by exploiting locally matched contact maps in a reference panel. By enabling high-resolution and robust analyses of topological domains from standard coverage Hi-C data, RobusTAD paves the way to gaining biological insights that had until now could only be possible from ultra-high coverage (and cost) data.

## Methods

### Notations

Consider an intra-chromosomal contact map $M = \{m_{ij}\}$, where $m_{ij}$ represents (normalized) interaction frequency between bin $i$ and $j$ at fixed resolution $r$. RobusTAD aims to detect TAD boundaries $B = \{B^L, B^R\}$ and TADs $D = \{(B^L_i, B^R_{i'})\}$, where $B^L$ and $B^R$ are lists of left and right boundaries respectively, $(B^L_i, B^R_{i'})$ indicates that the $i^{th}$ left and $i'^{th}$ right boundaries form a TAD. Define $M_{[a,b]}$ as the submatrix corresponding genomic region $[a, b]$, and $S_{[a,b]}$, $S^L_{[a,b]}$, and $S^R_{[a,b]}$ as the domain score, left and right boundary scores for TAD $(a, b)$.

### Boundary and domain scores

To calculate domain and boundary scores for $(a, b)$, we compared interactions for bins within $[a, b]$ and between bins in $[a, b]$ and its left and right flanking regions. Our null hypothesis in the nonparametric test assumes that, for each diagonal of the contact matrix, there are no differences between the distribution of within-TAD and across-TAD-boundary interactions (Additional file 1: Fig. S13). To quantify TAD, we performed a distance stratified (i.e., diagonal-wise) rank sum test between the two types of interactions. Define $D_k(a, b) = \{(i, i + k) : i \geq a, i + k \leq b\}$. We denote the TAD score evaluated from the $k^{th}$ diagonal as $S^k_{[a,b]}$, and compute it using within-stratum ranks as follows:

$$S^k_{[a,b]} = \frac{\sum_{(i,j) \in D_k(a,b)} \sum_{(i',j') \in D_k(a-k,a+k-1) \cup D_k(b-k+1,b+k)} \mathbb{1}(m_{i,j} > \gamma m_{i',j'}) - \mathbb{1}(m_{i,j} < \frac{1}{\gamma} m_{i',j'})}{\sum_{(i,j) \in D_k(a,b)} \sum_{(i',j') \in D_k(a-k,a+k-1) \cup D_k(b-k+1,b+k)} \mathbb{1}(m_{i,j} > \gamma m_{i',j'}) + \mathbb{1}(m_{i,j} < \frac{1}{\gamma} m_{i',j'})}$$

where $\gamma \geq 1$ controls the minimum gap allowed between the two types of interactions. Setting $\gamma = 1$ is equivalent to the Wilcoxon rank sum test. We set $\gamma = 1$, and found that values in the range of $[1, 1.3]$ tend to produce similar results (Additional file 1: Fig. S14). The overall TAD score for region $[a, b]$ is a weighted sum of the per-stratum scores:

$$S_{[a,b]} = \frac{2}{3(1+b-a)(b-a)} \sum_{k=1}^{k=b-a} (b-a+k+1)S_{[a,b]}^k$$

$b - a + k + 1$ is the number of Hi-C contact map entries used for comparison on the $k^{th}$ diagonal, $\frac{3(1+b-a)(b-a)}{2}$ is the total number of Hi-C contact map entries used for comparison. $S_{[a,b]}$ falls between $-1$ and 1, where $S_{[a,b]} = -1$ indicates all interactions inside the TAD are smaller than all interactions across TAD boundaries by a factor at least $\frac{1}{\gamma}$, $S_{[a,b]} = 0$ indicates no difference existed between the two types of interactions, $S_{[a,b]} = 1$ indicates all interactions inside the TAD exceed all interactions across TAD boundaries by a factor of at least $\gamma$. Note that if a nested TAD $(a', b')$ was already determined to occur within $(a, b)$ (with $a \leq a' < b' \leq b$), we exclude interactions belonging to $(a', b')$ from the calculation.

We define per-stratum left and right boundary scores $S_{[a,b]}^{Lk}$ and $S_{[a,b]}^{Rk}$, left and right boundary scores $S_{[a,b]}^{L}$ and $S_{[a,b]}^{R}$ similarly but only using interactions across the corresponding boundary as the background:

$$S_{[a,b]}^{Lk} = \frac{\sum_{(i,j)\in D_k(a,b)} \sum_{(i',j')\in D_k(a-k,a+k-1)} \mathbb{1}(m_{i,j} > \gamma m_{i',j'}) - \mathbb{1}(m_{i,j} < \frac{1}{\gamma}m_{i',j'})}{\sum_{(i,j)\in D_k(a,b)} \sum_{(i',j')\in D_k(a-k,a+k-1)} \mathbb{1}(m_{i,j} > \gamma m_{i',j'}) + \mathbb{1}(m_{i,j} < \frac{1}{\gamma}m_{i',j'})}$$

$$S_{[a,b]}^{Rk} = \frac{\sum_{(i,j)\in D_k(a,b)} \sum_{(i',j')\in D_k(b-k+1,b+k)} \mathbb{1}(m_{i,j} > \gamma m_{i',j'}) - \mathbb{1}(m_{i,j} < \frac{1}{\gamma}m_{i',j'})}{\sum_{(i,j)\in D_k(a,b)} \sum_{(i',j')\in D_k(b-k+1,b+k)} \mathbb{1}(m_{i,j} > \gamma m_{i',j'}) + \mathbb{1}(m_{i,j} < \frac{1}{\gamma}m_{i',j'})}$$

$$S_{[a,b]}^{L} = \frac{1}{(b-a)} \sum_{k=1}^{k=b-a} S_{[a,b]}^{Lk}$$

$$S_{[a,b]}^{R} = \frac{1}{(b-a)} \sum_{k=1}^{k=b-a} S_{[a,b]}^{Rk}$$

### Identifying candidate TAD boundaries

To identify domain boundaries from a normalized intra chromosomal contact map, we first compute a left boundary score $L_a = \max_{w \in \{w_{min},...,w_{max}\}} S_{[a,a+w]}^{L}$ for each bin $a$ along the whole chromosome. Right boundary scores are computed similarly: $R_b = \max_{w \in \{w_{min},...,w_{max}\}} S_{[b-w,b]}^{R}$. On one hand, $w$ should be as large as possible to include more interactions, thereby achieving a more robust estimation. On the other hand, $w$ should not exceed the size of the TAD being evaluated, as interactions beyond the TAD size would distort the boundary strength calculation. For our analyses at 5 kb resolution, we use $w_{min} = 50$ kb, and $w_{max} = 250$ kb. We also conducted the calculation with alternative settings and found that setting $w_{min}$ below 50 kb (i.e., 10 bins) often introduces false negative errors. In contrast, adjusting $w_{max}$ to other larger values has a negligible impact on the results.

To identify left and right boundaries from boundary scores, we use find_peak function in SciPy [45] to identify local peaks. We assume the minimum distance between two domain

boundaries is 25 kb (i.e., five 5-kb bins) and set *distance= 5* in find_peak. The set of putative boundaries this identifies usually contains false positives. We use FDR control to select a subset of high confidence boundaries. Briefly, we produced a decoy contact map by shuffling interactions diagonal-wise. The shuffling strategy destroys all domains but maintains the interaction frequency decay pattern. We identify domain boundaries from this decoy contact map and compare scores for boundaries identified in the original and decoy Hi-C contact maps. We select the top boundaries at a FDR of $\alpha$ ($\alpha = 0.05$ for data containing more than 300M valid read pairs, $\alpha = 0.1$ for data containing less than 300M valid read pairs).

### Refining boundary annotation by identifying locally matched chromosome conformations from the reference panel

Putative TAD boundaries predicted by the single-sample non-parametric test described above are often off by one or more bins, due to the noisy nature of the data. For a given left or right putative TAD boundary predicted at $b_i$, we define LMCCs as the subset of the Hi-C samples from our reference panel that have a predicted boundary with 25 kb (5 bins) of $b_i$. We then update the study sample's boundary scores for the 10-bin region centered at $b_i$ as the mean boundary scores of the study sample and all selected reference samples (Additional file 1: Note S2). We also provide users the option to compute refined boundary scores as a weighted sum of each sample's boundary scores in RobusTAD, with the weights set according to each sample's read depth. This approach slightly improved accuracy in our experiments. Last, we update the boundary call as the peak position among the refined boundary scores.

### Assembly of nested TADs from predicted boundaries

Given an intra-chromosomal contact map $M = \{m_{ij}\}$ and sets $B^L$ and $B^R$ of previously identified left and right boundaries, we sought to pair left and right boundaries to form hierarchical domains. Similar to OnTAD [15], we do not allow partial overlaps between domains in TAD predictions. This fully nested TAD hierarchy assumption allows us to use a dynamic programming algorithm to find a globally optimal solution. Our dynamic programming algorithm is inspired by the Nussinov algorithm [35] for RNA secondary structure prediction.

We denote the ordered multi-set of TAD boundaries as $(b_1, b_2, \ldots, b_{n-1}, b_n)$, where $b_i \in B^L \cup B^R$. We define the globally optimal solution as the nested TAD hierarchy that maximizes the sum of scores of all TADs in the hierarchy, subject to the TADs' left and right boundaries being selected (potentially with repetition) from $B^L$ and $B^R$. We create the dynamic programming table $T$ of size $n \times n$, where $T_{ij}$ stores the maximum sum of domain scores for all nested domains within region $[b_i, b_j]$. The forward pass of the dynamic programming fills the upper triangular portion of $T$, using the following recursion:

$$T_{ij} = \max_{i<k<j} T_{ik} + T_{kj} + \delta(i,j)$$

$$\delta(i,j) = \begin{cases} S_{[b_i,b_j]} & \text{if } b_i \in B^L \text{ and } b_j \in B^R \text{ and } S_{[b_i,b_j]} \geq \lambda \\ 0 & \text{otherwise} \end{cases}$$

$\lambda$ defines the minimum score for pairing $b_i$ and $b_j$ as a domain. Since the TAD score is defined similarly to the TAD boundary score, we set $\lambda = 0.2$, which approximates the minimum TAD boundary scores observed in most of our experiments. In addition, we found that setting $\lambda$ too high may result in fewer TAD predictions, but values in the range of [0, 0.3] perform well in practice (Additional file 1: Fig. S15). The evaluation of $\delta(i, j)$ depends on $k$ in the recursion function of $T_{ij}$ as $S_{[b_i, b_j]}$ requires excluding nested TADs within $[b_i, b_j]$ which are identified with the dynamic programming algorithm in previous steps. We sequentially fill entries in $T$ from the first to the furthest diagonals. To start, we initialize the first upper diagonal as

$$T_{i,i+1} = \begin{cases} S_{[b_i, b_{i+1}]} & b_i \in B^L, b_{i+1} \in B^R, \text{ and } S_{[b_i, b_{i+1}]} \geq \lambda \\ 0 & \text{otherwise} \end{cases}$$

Last, we select the optimal set of domains that maximize the sum of TAD scores for genomic region $[b_1, b_n]$ by backtracking from $T_{1n}$. The time complexity of this dynamic programming algorithm is $O(n^3)$ (where n is the number of boundaries) for filling the scoring table if $\delta(i, j)$ can be evaluated in constant time. However, since the evaluation of $\delta(i, j)$ involves comparing two sets of values that depend on $i$ and $j$, this comparison could take up to $O(n)$ time in the worst case. As a result, the overall time complexity of the algorithm could increase to $O(n^4)$. RobusTAD can be executed on a typical human data set at 5 kb resolution in about 1 day. Some domain boundaries in $b$ may be absent from the TAD hierarchy. They are treated as false positive domain boundaries or involved in partial overlap domains, which does not satisfy our assumption.

### Curating Hi-C reference panel

We downloaded 177 published human Hi-C datasets (Additional file 1: Table S1) from the GEO database and uniformly processed them with distiller [46]. Reads were mapped against hg38 and we discarded reads with a mapping quality < 10. This produced Hi-C contact maps at fixed resolutions and stored processed contact maps in multi-resolution cooler format (.mcool). Lastly, read count matrices were normalized using Cooler's iterative correction algorithm [38, 39]. We applied the single-sample version of RobusTAD with default parameters to calculate boundary scores for all of these Hi-C samples at 5 kb resolution and saved boundary scores and boundary calls as a reference database, to be used for the reference-panel based version of RobusTAD. This reference panel is specific to the data resolution. Applying RobusTAD to annotate Hi-C data at a different resolution requires users to create a corresponding reference panel, using the tools provided in the package.

### Enrichment analysis and Measure of Concordance

We followed Zufferey et al. [24] to analyze enrichment of H3 K36 me3 and H3 K27 me3 histone marks and CTCF, SMC3, and RAD21 structural proteins within TADs or at their boundaries. For structural protein enrichment, we calculated the fold-change by comparing peak counts in a narrow interval around a boundary to those in distant flanks (Fold change=$\frac{\text{peak}}{\text{background}} - 1$). For histone marks, we calculated the average log10-ratio in small intervals within TADs and obtained empirical $p$-values using ten shuffles.

To compare TAD partitions, we used the Measure of Concordance (MoC) [24], which ranges from 0 (absence of concordance) to 1 (full concordance) and is defined as follows,

$$\text{MoC}(\mathbf{P}, \mathbf{Q}) = \begin{cases} 1 & \text{if } N_P = N_Q = 1 \\ \frac{1}{\sqrt{N_P N_Q} - 1}(\sum_{i=1}^{N_P} \sum_{j=1}^{N_Q} \frac{|\mathbf{F_{i,j}}|^2}{|\mathbf{P_i}||\mathbf{Q_j}|} - 1) & \text{otherwise} \end{cases}$$

where $\mathbf{P} = \{\mathbf{P_i}\}$, and $\mathbf{Q} = \{\mathbf{Q_i}\}$ are sets of TADs including $N_P$ and $N_Q$ TADs, $\mathbf{F_{i,j}}$ is the overlap region between $\mathbf{P_i}$ and $\mathbf{Q_j}$, and $|\cdot|$ represents cardinality. We only included TADs without any smaller TAD in this analysis.

### Alternative approaches

This study compared RobusTAD to 14 other TAD callers. We ran TopDom, Armatus, Arrowhead, EAST, CaTCH, Domaincall (DI), GMAP, ICFinder, and HiCSeg as suggested in [24]. As we performed the analysis at 5kb resolution, we have updated parameters related to resolutions accordingly. We ran HiTAD, RefHiC, and deDoc with their default settings. OnTAD: We set maxsz = 600 to allow OnTAD to detect TADs as large as 3 Mb. Grinch: following Lee and Roy [17], we detected TADs by setting the expected TAD length as 2 Mb, 1 Mb, and 500 Kb in three runs and combined all results. We observed tools such as Grinch reported invalid TAD annotations of length less than three bins and excluded these invalid annotations from further investigation. In addition, the convention of TAD definition often varies from tool to tool (with ±1 bin shift). We converted them to the convention used in RobusTAD (i.e., boundary refers to the start of the left/right furthermost bin inside the TAD).

### Supplementary Information

---

Additional file 1: PDF document containing supplementary notes, figures, and table.

Additional file 2: Peer review history.

---

**Acknowledgements**
The authors thank Dr. Yue Li and Dr. Jacek Majewski for useful discussions in this project.

**Review history**
The review history is available as Additional file 2.

**Peer review information**

**Authors' contributions**
Y.Z. and M.B. conceived the study. R.D. co-conceived and implemented domain boundary annotation, and wrote portions of the manuscript. Y.Z. implemented RobusTAD, performed data analyses, and wrote the manuscript. M.B. supervised the project and wrote the manuscript. All authors read and approved the final paper.

**Funding**
This work was funded by a Genome Quebec/Canada grant to M.B.. Y.Z. is supported by FRQNT Doctoral (B2X) Research Scholarships.

**Data availability**
RobusTAD is available at https://github.com/zhyanlin/RobusTAD, and https://zenodo.org/records/15011037 [47], under MIT license. All scripts and data required to reproduce figures and analyses are available at https://doi.org/10.5281/zenodo.8306238 [48].
 Hi-C data: All Hi-C data used in this project are from previous publications [3, 42, 49–88] and were downloaded from public repositories. Accession numbers are provided in Additional file 1: Table S1.

## Declarations

### Ethics approval and consent to participate
Not applicable.

### Competing interests
The authors declare that they have no competing interests.

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

## 