## [Additional file 2: Peer review history. · Genome Biology]

Review history

First round of review

Reviewer 1

This work presents a new TAD calling method named RobustTAD, which provides accurate and robust TAD annotations in a hierarchical manner. The key idea behind this method is to incorporate information from existing Hi-C databases (i.e., Hi-C reference panels) to refine the annotated boundaries of the study sample. The authors demonstrate the strong performance of their method using several metrics. Overall, I find their method very interesting, and the manuscript is well-written.

I have several comments:

Major:

1. Page 5, line 52, what is the number of valid read pairs for the highest coverage data set?
2. Page 6, line 36, I am not very sure that the claim "RobustTAD identified the most ChIA-PET supported TADs from both contact maps" is correct. According to Figure 5, only small portion of TADs are CTCF supported (even for RobustTAD), and only small portion of CTCF contact pairs are TAD boundaries.
3. Page 7, line 41, is it possible to use some statistics to quantify the differences in Figure S6b?
4. Page 9, line 11, what is the time complexity of the dynamic programming algorithm used in this work?
5. Page 10, line 12, "values" here is not well defined, therefore, it is unclear to me whether the weight term " $b-a+k+1$ " is correct, (For me, $b-a-k+1$ is more reasonable, but I might misunderstand the term.) and this weight term is not consistent with the weight term used in Note S1.
6. Page 10, line 24, it would be better to provide the explicit form of the boundary score definition.
7. How do the authors choose the value of gamma? Do the choices of hyper-parameters, gamma and lambda, significantly impact the performance of the algorithm?
8. The algorithm of this work is based on the assumption that domain boundary is relatively well conserved across cells, however, there are some evidence (such as "analysis of the structural variability of topologically associated domains as revealed by Hi-C") showing that there are significant variations in TAD sets even between replicate samples. Did the authors meet some cases where this assumption is not valid, and hence using the reference panel make the results worse?
9. The method is tested on the 5kb resolution. How to use this method to call TADs at different resolutions?

Minor:

1. Page 6, line 41, Figure 5(d,e) need to be referred here.
2. Page 11, line 58, quality ≤ 10 , typo
3. Figure S5, "TAD" redundant

Reviewer 2

The authors propose RobustTAD, a computational method for identifying topological domains (TADs) from Hi-C data. Although there have been many TAD callers proposed, RobustTAD improves upon existing methods by employing reference Hi-C data sets when annotating a target data set. This makes sense because Hi-C has relatively little variation between cell types, but is very expensive to perform at fine resolution. RobustTAD builds off the authors' previous method RefHiC, which uses the same trick, by adding the capability to identify nested TADs. The authors evaluated RobustTAD in comparison with existing TAD callers. RobustTAD and RefHiC outperform alternatives: e.g. in comparison to ChIA-PET data, they achieve performance similar to the top-performing Arrowhead algorithm while predicting many more TADs.

The authors' evaluations are thorough and in agreement with the field's consensus. The manuscript is easy to understand. The novelty and impact are moderate since RefHiC previously introduced the idea of employing

reference Hi-C. I have several comments, none of which are major.

Comments:

- I thought RefHiC couldn't identify TAD hierarchies, but Fig 2 lists it as having both level 0 and level 1+ TADs. Why is this? If RefHiC can do so, what is the main novelty of RobusTAD?
- Fig 6: There has been much work clustering regulatory elements on the basis of TF binding. Are different clusters identified when restricting to TAD boundaries? Do previous TFBS clusters have the same patterns at TAD boundaries?

Minor:

- It seems like reference Hi-C data sets are given equal weight when refining boundaries. Would it make more sense to weight by read depth?
- P5L53: "fig." > "Fig." (Also several other instances.)
- P11L58: "!" > ">"

--

Maxwell Libbrecht
Associate Professor
School of Computing Science, Simon Fraser University

Authors' response to reviewers

Reviewer #1: This work presents a new TAD calling method named RobusTAD, which provides accurate and robust TAD annotations in a hierarchical manner. The key idea behind this method is to incorporate information from existing Hi-C databases (i.e., Hi-C reference panels) to refine the annotated boundaries of the study sample. The authors demonstrate the strong performance of their method using several metrics. Overall, I find their method very interesting, and the manuscript is well-written.

→ Thanks for your overall appreciation of our work.

I have several comments:

Major:

1. Page 5, line 52, what is the number of valid read pairs for the highest coverage data set?

→ There are 4 billion valid read pairs. We've updated our manuscript accordingly.

2. Page 6, line 36, I am not very sure that the claim "RobusTAD identified the most ChIA-PET supported TADs from both contact maps" is correct. According to Figure 5, only small portion of TADs are CTCF supported (even for RobusTAD), and only small portion of CTCF contact pairs are TAD boundaries.

→ Thank you for your comment. We agree that, according to Figure 5, only a small portion of TADs are supported by CTCF. However, it remains true that RobusTAD identified more ChIA-PET supported TADs than alternative tools. These poor matches might stem from differences in the number of CTCF interactions captured by different technologies. We used CTCF ChIA-PET data as the ground truth for CTCF interactions (2,168 and 4,957 CTCF interactions are revealed by ChIA-PET data for K562 and IMR-90 respectively). Nevertheless, this "ground truth" is far from perfect, and the number of interactions might be influenced by experimental settings. This suggests that TADs classified as "ChIA-PET unsupported" may indeed be supported by CTCF interactions that are missing in the ChIA-PET data.

3. Page 7, line 41, is it possible to use some statistics to quantify the differences in Figure S6b?

→ Thanks for your suggestion, we have added a two-tailed p-value to quantify the differences.

4. Page 9, line 11, what is the time complexity of the dynamic programming algorithm used in this work?
→ We have revised our manuscript to discuss the time complexity. The time complexity of this dynamic programming algorithm is $O(n^3)$ (where n is the number of boundaries) for filling the scoring table if $\delta(i,j)$ can be evaluated in constant time. However, since the evaluation of $\delta(i,j)$ involves comparing two sets of values that depend on i and j , this comparison could take up to $O(n)$ time in the worst case. As a result, the overall time complexity of the algorithm could increase to $O(n^4)$. Nonetheless, the algorithm can be executed on a typical human data set at 5 kb resolution in only about 1 day, which is comparable to or faster than many other TAD callers that can produce TAD hierarchies.

5. Page 10, line 12, "values" here is not well defined, therefore, it is unclear to me whether the weight term " $b-a+k+1$ " is correct, (For me, $b-a-k+1$ is more reasonable, but I might misunderstand the term.) and this weight term is not consistent with the weight term used in Note S1.

→ We apologize for any confusion caused by the term 'values.' In this context, 'values' refers to the number of Hi-C contact map entries within a region of interest. We have revised our manuscript to make this clearer.

Regarding the weight term, we confirm that ' $b-a+k+1$ ' is the correct form. This term represents the number of Hi-C contact entries within the region of interest (i.e., three consecutive triangles: upstream cross-boundary region, TAD, and downstream cross-boundary region) along the k -th diagonal. We have added Fig. S13 to illustrate this.

We'd like to remind the reviewer that ' $b-a+k+1$ ' is the weight term for the TAD score, whereas Note S1 pertains to the TAD boundary score. These two types of scores are slightly different, and we have added the explicit form for the boundary score in our manuscript to clarify this distinction.

6. Page 10, line 24, it would be better to provide the explicit form of the boundary score definition.

→ Thanks for your suggestion. We have provided the explicit boundary score definition in our manuscript.

7. How do the authors choose the value of gamma? Do the choices of hyper-parameters, gamma and lambda, significantly impact the performance of the algorithm?

→ We apologize for missing gamma value in our manuscript. We set gamma to 1 and have now included this in our manuscript. We chose this value for simplicity, as it aligns with the stratified Wilcoxon rank sum test. We tested a range of gamma values and evaluated accuracy by comparing boundary annotations against CTCF data. Our results indicate that the algorithm is robust across different gamma settings (Fig. S14), with values such as 1.0, 1.1, and 1.2 producing similar outcomes.

As for lambda, it represents the minimum TAD score required for a region to be identified as a TAD. Since our method of computing TAD scores is analogous to that of boundary scores, we believe that the range of TAD scores should be similar to boundary scores. Therefore, we set lambda to 0.2, which corresponds to the minimum TAD boundary scores observed across most of our experiments. It is also possible to select lambda by FDR, but this would be very time consuming. We also compared different values for lambda and found that it is robust as long as the value is not significantly larger or smaller than most of the boundary scores (Fig. S15).

8. The algorithm of this work is based on the assumption that domain boundary is relatively well conserved across cells, however, there are some evidence (such as "analysis of the structural variability of topologically associated domains as revealed by Hi-C") showing that there are significant variations in TAD sets even between replicate samples. Did the authors meet some cases where this assumption is not valid, and hence using the reference panel make the results worse?

→ Thank you for those suggestions. We have revised our manuscript accordingly. Whether TAD boundaries are conserved or not remains a debate, as there are research findings in favor of both viewpoints. We believe TADs are conserved to some extent. To mitigate the improper combination of non-conserved TADs while leveraging conserved information, RobusTAD selectively aggregates

samples with similar TAD boundary candidates identified in the initial step (i.e, having a domain boundary identified within a given window around the study sample's domain boundary).

Due to the lack of ground truth for conserved TAD boundaries. We evaluate whether including a reference panel leads to a more accurate annotation instead of analyzing cases where the conservation assumption does not hold. As shown in Figs. S11 and S12, using a reference panel generally improve accuracy. Given these results, although our boundary score refinement might struggle when reference samples have other domain boundaries very close to those of the study sample, we still believe that RobusTAD improves TAD annotation in most cases.

9. The method is tested on the 5kb resolution. How to use this method to call TADs at different resolutions?

→ Applying our methods to handle data at different resolutions requires the users to prepare a resolution-specific reference panel. This can be performed by applying our individual sample based TAD boundary scoring method on a list of reference samples. The user can obtain Hi-C samples that we used to create our reference panel from GEO. Alternatively, 4DN data portal also provides sufficient data to create such a panel.

Minor:

1. Page 6, line 41, Figure 5(d,e) need to be referred here.
2. Page 11, line 58, quality ≤ 10 , typo
3. Figure S5, "TAD" redundant

→ Thanks for pointing out typos and missing references, we have revised our manuscript as suggested.

Reviewer #2: The authors propose RobusTAD, a computational method for identifying topological domains (TADs) from Hi-C data. Although there have been many TAD callers proposed, RobusTAD improves upon existing methods by employing reference Hi-C data sets when annotating a target data set. This makes sense because Hi-C has relatively little variation between cell types, but is very expensive to perform at fine resolution. RobusTAD builds off the authors' previous method RefHiC, which uses the same trick, by adding the capability to identify nested TADs. The authors evaluated RobusTAD in comparison with existing TAD callers. RobusTAD and RefHiC outperform alternatives: e.g. in comparison to ChIA-PET data, they achieve performance similar to the top-performing Arrowhead algorithm while predicting many more TADs.

The authors' evaluations are thorough and in agreement with the field's consensus. The manuscript is easy to understand. The novelty and impact are moderate since RefHiC previously introduced the idea of employing reference Hi-C. I have several comments, none of which are major.

→ Thanks for your overall appreciation of our work.

Comments:

- I thought RefHiC couldn't identify TAD hierarchies, but Fig 2 lists it as having both level 0 and level 1+ TADs. Why is this? If RefHiC can do so, what is the main novelty of RobusTAD?

→ It's a good point. We agree with you that RefHiC couldn't identify optimal TAD hierarchies, but its output can include TAD-within-TAD cases. When RefHiC greedily pairs left and nearest right boundary as a TAD, we allow a left boundary to be paired with multiple right boundaries, this can lead to multiple TADs sharing the same left boundary, thus resulting level 0 and level 1+ annotations.

- Fig 6: There has been much work clustering regulatory elements on the basis of TF binding. Are different clusters identified when restricting to TAD boundaries? Do previous TFBS clusters have the same patterns at TAD boundaries?

→ We have revised the subsection "RobustTAD reveals multiple types of TADs" in our manuscript accordingly. While there has been much research on clustering regulatory elements based on TF binding, many of these studies focus on chromatin state annotation with a single piece of DNA segment as input. In contrast, our approach involves clustering TFBS specifically at the left and right boundaries of the same TAD, allowing us to analyze regulatory elements in pairs. This paired analysis of boundary regions provides a unique perspective that is often missing in previous studies.

Minor:

- It seems like reference Hi-C data sets are given equal weight when refining boundaries. Would it make more sense to weight by read depth?

→ Thank you for your suggestion. We have implemented the proposed read depth weighting in our software and mentioned it in the manuscript. While this approach slightly improves annotation accuracy when evaluated against CTCF data, its overall impact on our results is minimal. Therefore, we have decided not to reproduce the primary analyses using this weighting scheme, as it does not significantly affect our study's conclusions.

- P5L53: "fig." > "Fig." (Also several other instances.)

- P11L58: "!" > ">"

→ Thanks for pointing out typos and missing references, we have revised our manuscript as suggested.

--

Second round of review

Reviewer 1

I would like to thank the authors for their detailed responses. According to the revised manuscript and the responses, I have the following comments:

1. If those two blue triangles in Fig. S13 is to represent $D_{\{k\}}(2a-b, a-1)$ and $D_{\{k\}}(b+1, 2b-a)$, I don't think they are correctly shown. Instead, the blue triangles should be the ones that are adjacent to the diagonal, like the green triangle. (Intuitively, according to the definition of $D_{\{k\}}(a,b)$, it is unlikely that we will have more entries when k increases).
2. Because of point 1, it is still unclear to me why the term $b-a+k+1$ is correct. Besides, since $S^{\{k\}}[a,b]$ is to compare pairs of entries, the authors may also need to consider whether to use quadratic (i.e. $(b-a+k+1)^2$) instead of linear.
3. Definitions of $S^{\{L\}}[a,b]$ and $S^{\{R\}}[a,b]$, since the term $b-a+1$ is not related to k , it is unclear to me why it is not canceled with the same term in the denominator.
4. Page 11, how to pick the values of $w_{\{\min\}}$ and $w_{\{\max\}}$?

5. Do the authors have evidence supporting the claim “...about 1 day, which is comparable to or faster than many other TAD callers that can produce TAD hierarchies”? According to my experience of using TAD callers, it is not the case.

Authors' response to reviewers

Reviewer #1: I would like to thank the authors for their detailed responses. According to the revised manuscript and the responses, I have the following comments:

→ We thank you for your overall appreciation of our work!

1. If those two blue triangles in Fig. S13 is to represent $D_{\{k\}}(2a-b, a-1)$ and $D_{\{k\}}(b+1, 2b-a)$, I don't think they are correctly shown. Instead, the blue triangles should be the ones that are adjacent to the diagonal, like the green triangle. (Intuitively, according to the definition of $D_{\{k\}}(a,b)$, it is unlikely that we will have more entries when k increases).

→ Thank you for pointing out the issue regarding our formula. We apologize for not addressing your concern in our first revision. Upon further reflection, we now agree with you that we made a mistake in our formula. The blue and green triangles represent the regions we compare in our algorithm, but their mathematical definitions are incorrect. We have now revised our formula. $D_{\{k\}}(2a-b, a-1)$ and $D_{\{k\}}(b+1, 2b-a)$ should be $D_{\{k\}}(a-k, a+k-1)$, and $D_{\{k\}}(b-k+1, b+k)$.

2. Because of point 1, it is still unclear to me why the term $b-a+k+1$ is correct. Besides, since $S^{\{k\}}[a,b]$ is to compare pairs of entries, the authors may also need to consider whether to use quadratic (i.e. $(b-a+k+1)^2$) instead of linear.

→ Thank you for your comment. As explained above, the term " $b-a+k+1$ " represents the number of entries along the k -diagonal that we use to perform pairwise comparisons, which corresponds to all the values included in that diagonal. During the development of our algorithm, we also experimented with the quadratic form, such as using " $(b-a+k+1)^2$ ", and weighting the scores based on the number of pairwise comparisons from each stratum. We found that both the trend of scores and the boundaries called were very similar across these variations. Given that " $b-a+k+1$ " is simple and also used in Devan et al. [1] to weight strata scores, we decided to adopt this term in our score calculation.

3. Definitions of $S^{\{L\}}[a,b]$ and $S^{\{R\}}[a,b]$, since the term $b-a+1$ is not related to k , it is unclear to me why it is not canceled with the same term in the denominator.

→ Good point, and we apologize for it. We have canceled out the term.

4. Page 11, how to pick the values of $w_{\{min\}}$ and $w_{\{max\}}$?

→ The parameter w is a hyperparameter used to evaluate TAD boundary strength. On one hand, w should be as large as possible to include more interactions, thereby achieving a more robust estimation. On the other hand, w should not exceed the size of the TAD being evaluated, as interactions beyond the TAD size would distort the boundary strength calculation.

Since the exact size of a TAD is unknown in advance, we rely on the observation that most TADs are larger than 50 kb. Based on this, we set $w_{\{min\}}=50$ kb as a starting point. To ensure robustness for larger TADs, we calculate boundary scores across a range of $w \in [w_{\{min\}}, w_{\{max\}}]$ and use the maximum score within this range as the final boundary strength. In our implementation, we set $w_{\{min\}}=50$ kb, and $w_{\{max\}}=250$ kb.

We also conducted the calculation with alternative settings and found that setting w_{\min} below 50 kb (i.e, 10 bins) often introduces false negative annotations. In contrast, adjusting w_{\max} to other larger values has a negligible impact on the results.

5. Do the authors have evidence supporting the claim "...about 1 day, which is comparable to or faster than many other TAD callers that can produce TAD hierarchies"? According to my experience of using TAD callers, it is not the case.

→ Thank you for your comment. In our experiments, RobusTAD completed the full genome analysis in approximately one day, with the annotation of TADs from a single chromosome taking about one hour. While RobusTAD is not included in the benchmarking table S3 in [2], the runtime of other tools in Table S3 demonstrates a range of performance levels, with five tools taking approximately one hour to analyze human chromosome 7. This suggests that RobusTAD's speed is reasonable for tools capable of producing TAD hierarchies. However, we acknowledge that this does not necessarily mean RobusTAD is faster than many other TAD callers. We have revised the manuscript accordingly.

Reference

1. Mehrotra, Devan V., Xiaomin Lu, and Xiaoming Li. "Rank-based analyses of stratified experiments: alternatives to the van Elteren test." *The American Statistician* 64.2 (2010): 121-130.

2. Xu, Jingxuan, et al. "A comprehensive benchmarking with interpretation and operational guidance for the hierarchy of topologically associating domains." *Nature Communications* 15.1 (2024): 4376.

Third round of review

Reviewer 1

I would like to thank the authors for their detailed responses. I don't have any other major comments. Only two typos:

1. At the beginning of page 11, $R_{\{a\}}$ should be R_b .
2. Figure S13, I think (b) is the right boundary score not left.